# Immune Response 5–7 Months after Vaccination against SARS-CoV-2 in Elderly Nursing Home Residents in the Czech Republic: Comparison of Three Vaccines

**DOI:** 10.3390/v14051086

**Published:** 2022-05-18

**Authors:** Jan Martínek, Hana Tomášková, Jaroslav Janošek, Hana Zelená, Alena Kloudová, Jakub Mrázek, Eduard Ježo, Vlastimil Král, Jitka Pohořská, Hana Šturcová, Rastislav Maďar

**Affiliations:** 1Institute of Public Health Ostrava, Partyzánské náměstí 7, 702 00 Ostrava, Czech Republic; jan.martinek@zuova.cz (J.M.); hana.zelena@zuova.cz (H.Z.); alena.kloudova@zuova.cz (A.K.); jakub.mrazek@zuova.cz (J.M.); eduard.jezo@zuova.cz (E.J.); 2Department of Epidemiology and Public Health, Faculty of Medicine, University of Ostrava, Syllabova 19, 703 00 Ostrava, Czech Republic; rastislav.madar@osu.cz; 3Centre for Health Research, Faculty of Medicine, University of Ostrava, Syllabova 19, 703 00 Ostrava, Czech Republic; jaroslav.janosek@osu.cz; 4Institute of Public Health Ústí nad Labem, Moskevská 1531/15, 400 01 Ústí nad Labem, Czech Republic; vlastimil.kral@zuusti.cz (V.K.); jitka.pohorska@zuusti.cz (J.P.); hana.sturcova@zuusti.cz (H.Š.)

**Keywords:** COVID-19, humoral immunity, cellular immunity, elderly nursing home residents

## Abstract

Background and Aims: Elderly nursing home residents are especially prone to a severe course of SARS-CoV-2 infection. In this study, we aimed to investigate the complex immune response after vaccination depending on the convalescence status and vaccine. Methods: Sampling took place in September–October 2021. IgG antibodies against spike protein and nucleocapsid protein, the titer of virus neutralization antibodies against delta and (on a subset of patients) omicron, and cellular immunity (interferon-gamma release assay) were tested in nursing home residents vaccinated with Pfizer, Moderna (both 30–31 weeks after the completion of vaccination), or AstraZeneca (23 weeks) vaccines. The prevalence with 95% confidence intervals (CI) was evaluated in Stata version 17. Results: 95.2% (95% CI: 92.5–97.1%) of the 375 participants had positive results of anti-S IgG, 92.8% (95% CI: 89.7–95.2%) were positive in virus neutralization assay against delta, and 89.0% (95% CI: 84.5–92.5%) in the interferon-gamma-releasing assay detecting cellular immunity. Results of the virus neutralization assay against omicron correlated with those against delta but the neutralization capacity was reduced by about half. As expected, the worst results were found for the AstraZeneca vaccine, although the vaccination-to-test period was the shortest for this vaccine. All immune parameters were significantly higher in convalescent residents than in naive residents after vaccination. No case of COVID-19 occurred during the vaccination-to-test period. Conclusions: A high immune response, especially among vaccinated convalescents (i.e., residents with hybrid immunity), was found in elderly nursing home residents 5–7 months after vaccination against SARS-CoV-2. In view of this, it appears that such residents are much better protected from COVID-19 than those who are only vaccinated and the matter of individual approach to the booster dose in such individuals should be further discussed.

## 1. Introduction

Vaccination against SARS-CoV-2 is currently a much-discussed topic both in academic and public spaces. The necessity for the population-wide application of booster doses is also subject to much controversy—while it is deemed necessary by some, others claim that this application should be individualized as, in many people, immunity acquired after the original vaccination or previous COVID-19 infection, and especially by hybrid immunity, is sufficient for maintaining long-term protection, if not against the infection then at least against a severe course characterized by ICU hospitalization or death, e.g., [1,2,3,4].

The two mechanisms of immunity, i.e., humoral and cellular immunity, play together a crucial role in the defence of the body against the virus. The humoral (antibody-mediated) immunity including IgM, IgA, and, especially, the long-term IgG antibodies is the easiest to measure and, to a certain extent, represents an indicator of the activity of B cells. However, it is important to note that even though the antibody levels wane over time, the memory B-cells remain in circulation for a long time, trained and ready to trigger antibody production at any time [5].

Unlike measurement of the individual types of antibodies, the virus neutralization test does not measure any specific class but rather a complex immune response of the humoral immunity. It, however, requires a biosafety level 3 for laboratories (BSL-3) and is laborious and time-demanding, which makes it unsuitable for routine use, especially as it has been shown that its results generally correlate well with IgG antibodies [6,7,8]. Nevertheless, there still may be individuals who, despite having low IgG levels, exert positivity in the virus neutralization test [9,10].

The term “cellular immunity” refers to T-cell-mediated immune protection. These cells participate in multiple processes, from direct elimination of infected cells through inhibition of viral replication and triggering of other pathways by interferon-gamma (IFN-γ) release to modulating humoral response [11]. In this respect, IFN-γ release, which is relatively easily measurable by IFN-γ-releasing assay (IGRA) [12], is generally considered a useful indicator of the activity of T-cells.

Elderly individuals, especially those living in nursing homes, are of particular concern as this group was among the most severely affected, especially in the first waves of the pandemic [13]. How long the protection attributed to vaccination lasts in this age group is, therefore, of great interest. Individuals above 80 years of age were also shown to develop lower cellular and humoral immunity after vaccination [14,15]. A study from the Czech Republic that took place in summer 2021 reported that the levels of IgG antibodies in the elderly population after vaccination were very low; however, that study used a rapid point-of-care test rather than a laboratory ELISA assay [16], which is less sensitive than ELISA determination.

For this reason, after a discussion with the Czech Ministry of Health, we performed an evaluation of the immune response in residents of nursing care homes vaccinated with two doses of various types of vaccines (Spikevax by Moderna, Cominarty by Pfizer, and Vaxzevria by AstraZeneca; throughout this text, manufacturer’s names will be used) using more sensitive methods. In particular, we aimed (i) to find out the level of humoral as well as cellular immune protection of elderly inhabitants of nursing homes in a large sample 7 months after vaccination with Moderna, (ii) to compare it with those vaccinated with Cominarty (also after approx. 7 months) and AstraZeneca (approx. 5 months), respectively, and (iii) to compare the immune response in elderly nursing home residents who were only vaccinated to the response in those who have, besides vaccination, also had the COVID-19 disease (not only on the basis of medical records but also on the basis of the presence of antibodies against SARS-CoV-2 nucleocapsid protein; this type of antibody is not produced after vaccination but rather only after SARS-CoV-2 infection). In addition, we aimed (iv) to compare on a subset of samples the difference in their virus neutralizing capacity against the delta and omicron variants of SARS-CoV-2.

## 2. Methods

### 2.1. Samples

Participant recruitment took place in six randomly selected nursing homes in two regions of the Czech Republic (Moravian-Silesian and Usti) that met the selection criteria: vaccination during January–May 2021 in 100 or more clients, history of positively tested COVID-19 patients in the nursing home, and consent of the facility management to participation in the study.

Inclusion criteria for individual participants were: age 65+, living in the respective facility at least since 1 January 2021, two doses of a vaccine against SARS-CoV-2 received in the period February–March 2021 (Moderna), March 2021 (Pfizer), and May 2021 (Astra Zeneca), patient’s consent to the participation in the study and willingness (and ability) to fill in the health questionnaire. The differences in the period of vaccination were caused by the availability of vaccines at the time. In the health questionnaire, all participants answered questions related to basic demographic data (age and sex), vaccination (dates and vaccine), personal history of COVID-19 (including the necessity for hospitalization due to COVID-19), comorbidities (diabetes mellitus, cardiovascular diseases, hypertension, respiratory diseases, oncological/hematooncological diseases, obesity with BMI > 30 kg m^−2^, or other diseases), and usage of drugs affecting immunity (corticosteroids, methotrexate, biological treatment, etc.). Individual clients were anonymized using a barcode. From each client, 5 mL of venous blood was collected for humoral immunity testing, and 2 mL was collected into a separate tube (heparinized blood) for testing of cellular immunity. Sampling took place in September–October 2021. All samples were stored at −20 °C. Later, 40 samples were selected based on patients’ COVID-19 history evaluated according to the medical records and the presence of anti-N IgG (10 samples from each combination of positivity/negativity of anti-N IgG and history of positive PCR test, see below); we also took into account the proportion of vaccine types in individual groups and selected the samples in a way corresponding to this proportion in individual groups.

### 2.2. Evaluation of Humoral Immunity

IgG antibodies against the S protein (anti-S IgG) were semiquantitatively determined by ELISA using the kit Anti-SARS-CoV-2 Quantivac ELISA IgG (anti S1) (Euroimmun, Lubeck, Germany) in accordance with the manufacturer’s instructions. As all participants in our resident group were vaccinated, the presence of antibodies against the spike protein (anti-S antibodies) could not be used as proof of the personal history of infection. For this purpose, we performed an additional analysis of antibodies against the nucleocapsid protein (anti-N IgG) to determine whether or not the participants had COVID-19 in the past (independently on the medical records) using the kit Anti-SARS-CoV-2 Quantivac ELISA IgG (anti NP) (Euroimmun, Lubeck, Germany) in accordance with the manufacturer’s instructions. The values were expressed as positivity index (PI, also known as signal to cut-off ratio; calculated as the ratio between the sample response and the response of the internal cut-off control; PI values < 0.9 indicate a negative result, 0.9–1.1 a borderline result, and PI over 1.1 indicates the presence of antibodies in the sample).

A virus neutralization test indicating the total humoral immune response was performed in sterile 96-well plates as described in detail before [17]. In brief, the SARS-CoV-2 strain (wild-type) and CV-1 cells (African green monkey kidney fibroblasts) were used for testing. Serum samples were diluted in two-fold dilutions which, after mixing with the virus, resulted in final serum concentrations of 1/10, 1/20 … up to 1/2560. These dilutions were mixed with the virus solution (100 infection doses) and incubated overnight at +4 °C. The next day, 25 µL of CV-1 cell suspension was added into each well and the plates were incubated for further 3–4 days at 37 °C and 5% CO_2_ atmosphere. After that, 25 µL of neutral red dye (1:10,000 aqueous solution) was added into each well and the mixture was incubated for additional 24 h under the same conditions. Only live uninfected cells were stained with the neutral red dye, enabling macroscopic reading. The virus neutralization titer was determined as the inverted value of the highest dilution of the sample neutralizing the cytopathic effect of the virus by more than 50%. Values below 10 were considered proof of absence of any humoral immunity against SARS-CoV-2 and values of ≥20 were considered positive. The virus neutralization titer was evaluated against delta and (on a subset of patients) omicron strains.

### 2.3. Evaluation of Cellular Immunity

Cellular immunity was determined by the interferon-γ release assay (IGRA). This test is based on the ability of lymphocytes (activated Th1, cytotoxic T lymphocytes, and natural killer—NK—cells) stimulated by specific antigens or mitogens to produce interferon gamma (IFN-γ). In this test, 0.5 mL of heparinized blood is pipetted into special antigen-containing incubation tubes (SARS-CoV-2 IGRA stimulation tube set, ET 2606-3003; Euroimmun, Lubeck, Germany) and incubated for 20–24 h, which leads to stimulation of the T and NK cells. Subsequently, the blood is centrifuged (12,000 g) and the levels of produced IFN-γ levels measured using ELISA kits (product EQ 6841-9601; Euroimmun, Lubeck, Germany). The IGRA tests were classified as positive where values of >200 mIU/mL were recorded, borderline (100–200 mIU/mL), or negative < 100 (mIU/mL).

### 2.4. Statistical Analysis

Prior to the study, the sample size necessary for the evaluation of the differences in immune response between vaccines was estimated. The estimated sample size was calculated with assumptions of 50 ± 5% seropositivity and a population of 20,000 (approximate number of elderly people 65+ in the two regions participating in the study).

The prevalence of antibodies (IgG and VNT) was evaluated both qualitatively and quantitatively. Standard methods of descriptive statistics were used; differences between groups were tested using Mann–Whitney (two groups) or Kruskal–Wallis (more than two groups), and Dunn’s tests, chi-squared test or Fisher exact test. The correlation between results was measured by Spearman correlation coefficient (r_s_). All testing was performed at the 5% level of significance in Stata, v. 17.

The study was approved by the Ethics Committee at the National Institute of Public Health Ostrava, No. P02/2021.

## 3. Results

In all, 375 inhabitants of nursing homes were included in the study. Of these, 209 (56%) received the Moderna vaccine (median period from the second dose to sampling of 215 days), 105 (28%) received the Pfizer vaccine (207 days), and 61 (16%) were vaccinated with the AstraZeneca vaccine (159 days). Details on the individual populations, including the results of the N-protein analysis indicating the history of COVID-19 independently of the medical records, are shown in Table 1.

None of the study participants was infected with COVID-19 in the period between the administration of the second dose of the vaccine and testing (however, it must be noted that the COVID-19 prevalence in the population at the time when the immune protection was likely to wane—summer 2021—was generally low).

### 3.1. Humoral Immunity

In total, 357 participants (95.2%; 95% CI: 92.5–97.1%) of elderly individuals in the nursing homes had a positive result of anti-S IgG antibodies. The highest representation of negative results was among residents who received the Astra Zeneca vaccine; the differences were, however, statistically borderline insignificant (Fisher’s exact test, *p* = 0.056). Figure 1A details the distribution of the anti-S IgG positivity index in elderly residents vaccinated with all three vaccines, from which the higher relative representation of low values in AstraZeneca (despite the significantly shorter vaccination-to-test period) is apparent.

These results correlated well with those of the virus neutralization test against the delta strain, which were positive in 92.8% of all study participants (Figure 1B, Table 2), with an additional 2.4% of participants having a borderline result. Again, the highest representation of negative results was among participants vaccinated with AstraZeneca, followed by Pfizer; the highest immune response was stimulated by the Moderna vaccine. The differences between groups were statistically highly significant (Fisher’s exact test, *p* < 0.001).

Cellular immunity was examined only in a subset of 254 participants, with 14 (5.5%) individuals showing negative values, the same number borderline values, and 89% of examined participants were positive (see Table 2). No statistically significant differences were found among groups (Fisher’s exact test, *p* = 0.103).

Evaluation of the association between anti-S IgG and VNT confirmed the expected—a statistically significant association was identified between these two variables (Spearman correlation coefficient of r_s_ = 0.82; *p* < 0.001) (Figure 2A). Similarly, highly significant correlations were identified also between anti-S IgG and T-cell-mediated immunity (r_s_ = 0.622; *p* < 0.001) and VNT vs. T-cell immunity (r_s_ = 0.644, *p* < 0.001), respectively (Figure 2B,C).

### 3.2. Immune Response in Association with the Personal History of COVID-19

Two indicators of the personal history of COVID-19 were used for evaluating its effect on the immune response in vaccinated nursing home residents: medical records describing the history of PCR positivity for SARS-CoV-2 and, as an independent objective parameter, anti-N IgG. Anti-N IgG antibodies were detected in 162 (43.2%) study participants (range among borderline and positive 0.90–4.85).

Recipients of each vaccine were classified into four groups according to the medical records and anti-N IgG (see Figure 3):Group 1: No COVID-19 record in personal history/no anti-N IgGGroup 2: COVID-19 record in personal history/no anti-N IgGGroup 3: No COVID-19 record in personal history/anti-N IgG positiveGroup 4: COVID-19 record in personal history/anti-N IgG positive

As obvious from Figure 3, Groups 1 and 4 (i.e., groups in which the anti-N IgG corresponded with medical records) constituted the majority of cases (approximately two-thirds of all cases in each group). There were obvious differences between the distribution into groups between individual vaccines (chi-squared test, *p* < 0.001). In the AstraZeneca group, no individuals who had COVID-19 and, at the same time, did not have the anti-N IgG, were detected. On the contrary, 69 (18.7%) study participants were anti-N IgG positive (i.e., had COVID-19 previously) but their medical records did not show this information. It is, therefore, reasonable to assume that the infection was asymptomatic in these individuals.

Figure 4 shows the combined (total across all vaccines) distributions of the anti-S IgG. We can clearly see that the distributions differ across the groups (*p* < 0.001, Kruskal–Wallis test). Post hoc analysis detected statistically significant differences among all groups (*p* < 0.01, Dunn’s test) with the exception of the two groups with anti-N IgG antibodies (*p* = 0.475). These latter two groups were associated with the highest ani-S IgG values regardless of the medical history records. Contrary, the group with no medical history of COVID-19 and no anti-N IgG showed significantly lower levels of anti-S IgG than the remaining groups. Due to the high correlation of anti-S IgG and VNT, the relationship of individual groups to VNT is not presented here.

In the groups of residents with a history of COVID-19 (be it based on medical records or anti-N antibodies), only three participants in total had a negative result of T-cell-mediated immunity. Only five individuals in the sample showed negative results in all studied parameters (IgG, VNT, and IGRA); all these participants suffered from multiple chronic conditions, namely diabetes (*n* = 4), cardiovascular diseases (*n* = 5), and hypertension (*n* = 5); one of these residents had, in addition, a hematooncological disease.

Comparison of the virus neutralization titer against delta and omicron on a subset of 40 patients revealed a good correlation, with Spearman correlation coefficient of 0.874. The neutralization capacity of the samples against omicron was, however, about half that of the capacity against delta (see Table 3). The difference was statistically significant (*p* < 0.001). In 6 samples out of the subset of 40 (15%), the neutralizing capacity was equal for both strains, higher for omicron than for delta; in 2 samples, the virus neutralization capacity was higher for omicron than for delta and the remaining 32 samples showed lower neutralizing capacity against omicron than against delta.

## 4. Discussion

The need for the booster dose of a vaccine after several months has been subject to much discussion in both the scientific community and the media worldwide. The immune protection from infection as well as from a severe course or death have been repeatedly shown to wane over several months (e.g., [18,19]). Still, the vaccine effectiveness against hospitalization or death was consistently higher than against infection [18,20]. This, theoretically, might be associated with the fact that vaccination does not confer mucosal immunity. We can also hypothesize that in a part of the vaccinated population, the immune response to the exposure to the virus (i.e., the humoral and cellular immunity resulting from vaccination) might be insufficient for preventing infection but as the immune system is still “pre-activated”, its fast reaction can prevent a serious course of the disease.

Identification of individuals whose stimulation of the immune system by vaccination wanes quickly or who have not responded to vaccination at all is extremely important for their own protection as the false sense of security provided by their vaccination status may lead to the increase of their risky behaviour and to contagion (and further spreading) of the disease. In our study, only 4.8% of individuals were found to have no (or borderline) IgG and 7.2% to have negative or borderline VNT results after approx. 7 (Pfizer and Moderna) and 5 (AstraZeneca) months, respectively. Of these, the highest representation of negative or borderline immune response was recorded for the AstraZeneca vaccine (9.9% of IgG negative and almost 20% of VNT negative or borderline results, respectively), although the recipients of this vaccine were tested after a significantly shorter period of time compared to the mRNA vaccines. Of those, Moderna yielded better results than Pfizer in the humoral immune response, with only 2.8% of individuals vaccinated with Moderna having negative or borderline VNT test. These results are in accordance with real-life evaluations of the protection against disease reported for individual vaccines in the literature, e.g., [18,19,21].

Interestingly, the situation is different where cellular immunity is evaluated. Although no statistically significant differences between individual vaccines were detected at the level of *p* < 0.05, the Pfizer vaccine appears to have the best results where the proportion of individuals with positive results of the IGRA assay is concerned. This can, however, be associated with the fact that the proportion of individuals with a history of COVID-19 (determined by either medical records or the presence of anti-N IgG) was the highest in the Pfizer group.

The results of anti-S IgG highly correlated with the results of the virus neutralization test. This also means that the simple determination of anti-S IgG can be used as a good proxy of virus-neutralizing capacity and, thus, of immune protection [8,22]. As expected, the virus neutralization capacity of most samples against the omicron strain of SARS-CoV-2 was lower than against the delta strain; this was especially true about samples with higher VNT capacity against delta (from 40 onwards) while for samples that were borderline positive or negative even for delta (VNT titers of 20 and less), the results were in agreement. Still, we must take into account that this analysis was only performed on a small subset of samples and a more detailed analysis would be needed for any strong statements.

The fact that all groups with COVID-19 history (regardless of whether this was confirmed based on a PCR-positive test or the N-protein) had higher median values of IgG is in accordance with the literature (see Figure 4; as the IgG, VNT, and IGRA highly correlated, only the figure for IgG is presented) [4,23]. As both figures indicate, the humoral immune response in Group 2 (residents with positive PCR test but no anti-N IgG) is much more variable than the groups with positive anti-N IgG. A possible explanation for this is discussed in our previous paper dealing with PCR and rapid antigen testing in association with testing for the presence of viable virus [24]; in some individuals who were positively PCR-tested, the test might have detected only viral debris (RNA of inactive virus—for example, viral particles that have already reached the mucosa in an inactivated state or that were inactivated by non-specific mucosal immunity) and such persons might actually have not developed humoral/specific cellular immunity. This group is, therefore, likely to contain a mix of individuals who really were and were not infected with SARS-CoV-2, which is reflected in the high variability of the humoral response in this group.

Several studies have reported similar results. In residents without previous COVID-19 infection, Blain et al. [4] found only 3% of seronegative elderly residents 6 weeks after the second dose of the vaccine. The authors also confirmed that the highest levels of anti-RBD antibodies were found in individuals who had COVID-19 prior to vaccination (as can be clearly seen also in our results as well as in those by Pannus et al. [25]) and, in effect, recommended that only a single dose of a vaccine should be applied to participants with COVID-19 history. Kontopoulou et al. [26] also reported a 99% seropositivity in elderly individuals 3 weeks after vaccination. On the other hand, Witkowski et al. [15] found that 11.6% of nursing home residents did not seroconvert after two doses of the Pfizer vaccine and their average neutralizing activity was significantly lower than in younger individuals (workers from the same nursing home).

The protection conferred by humoral immunity remains a much-discussed topic and many countries, including the Czech Republic, do not consider even the highest levels of IgG antibodies as protective. Although thresholds for virus neutralization titers for preventing SARS-CoV-2 infection have not yet been determined, several papers on this topic have been published already and high levels of neutralizing antibodies have been shown to protect from the disease. In their study focusing on an outbreak of COVID-19 within a nursing home approx. 6 months after vaccination, Pierobon et al. [22] reported that patients with anti-S IgG levels over 50 BAU/mL (which corresponds approx. to the positivity index of 1.4 in our study) were at about five times lower risk of a severe COVID-19 course than those with levels < 50 BAU/mL; none of the residents with IgG > 1000 BAU/mL had a serious course of COVID-19.

Havervall et al. [2] reported the approx. 100 times lower risk of infection in seropositive COVID-19 convalescent individuals compared to seronegative ones. Although their paper did not discuss the association of the presence of antibodies among vaccinated individuals, it can be considered an indication of the protective immunity posed by the presence of these antibodies and cellular immunity.

Together, these results bring a question of how many of these residents do indeed need a booster dose. Khopury et al. [27] suggested that neutralizing titer corresponding to 20% of the mean neutralization activity detected in convalescent individuals in the respective laboratory provides 50% protection from the contagion of COVID-19; where severe COVID-19 is concerned, the protective level was shown to be as low as 3% of the mean convalescent neutralizing activity. More importantly, graphs shown in their paper (as well as simple logic) indicate that the virus neutralization capacity equal to that of convalescents is associated with more than 80% protection from any COVID-19 and, thus almost 100% protection from severe disease. Based on our long-term database from another study that is underway, the median neutralization titer in COVID-19 residents without previous vaccination or COVID-19 history 1–2 months after convalescence in our settings is 160 (unpublished data). As the neutralizing titers exceeded this threshold in 45% of residents who were only vaccinated but had no previous history of COVID-19 while it was present in 94% of individuals with a history of COVID-19, it appears that few individuals in the latter group would likely benefit from a booster vaccination. Although adverse effects after the booster dose are generally considered rare [28], they should not be taken lightly and unnecessary vaccination of individuals with already highly stimulated immunity is not likely to improve their immunity against SARS-CoV-2. For this reason, the matter of individual approach to the booster dose in such individuals should be further discussed [29,30,31].

### Limitations of the Study

The fact that individuals after an AstraZeneca vaccine were tested sooner after vaccination than those vaccinated with Pfizer and Moderna is an obvious limitation of the presented study. However, we can see that even after this shorter period of time, AstraZeneca yielded the poorest results; they would be likely even worse 7–8 weeks later to be on par with mRNA vaccines but the indication of its lowest immunogenetic activity known from other studies (e.g., [19]) was confirmed.

It would be also interesting to observe the occurrence of the disease during the autumn (delta) and winter (omicron) waves; this was impossible as a vast majority of the study participants were vaccinated with booster doses.

## 5. Conclusions

We found that the immune response in elderly nursing home residents after full vaccination was maintained in 95.2% (anti-S IgG), 92.8% (virus neutralization test), and 89.0% (interferon-gamma-releasing assay) of tested individuals; 23 (AstraZeneca), 30 (Pfizer), or 31 (Moderna) weeks, respectively, after the completion of vaccination. Despite the shortest vaccination-to-test period, the highest representation of negative results was recorded for the AstraZeneca vaccine. Most importantly, we found that in individuals who had COVID-19 previously (i.e., those with hybrid immunity), a robust and high immune response persisted, significantly higher than in individuals who were SARS-CoV-2 naive before vaccination. In view of this, we suggest that such residents are much better protected from COVID-19 than those who are only vaccinated and the matter of individual approach to the booster dose should be further discussed. The results of a sub-analysis comparing virus neutralization capacity against the delta and omicron strains showed that although there was a good correlation in VNT results against these two strains, the neutralizing capacity against omicron was reduced by about 50%.

## Figures and Tables

**Figure 1 viruses-14-01086-f001:**
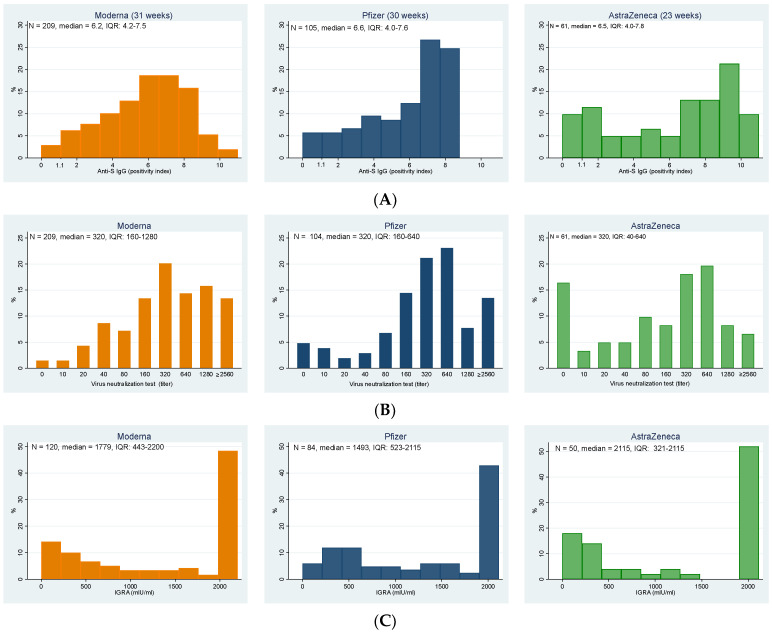
Distributions of the (**A**) levels of anti-S IgG, (**B**) titers of virus neutralization test against delta strain, and (**C**) results of interferon-gamma release assay indicating T-cell immunity for individual vaccines; note that the vaccination-to-test period for AstraZeneca is shorter (23 weeks) than for the Moderna and Pfizer (31 and 30 weeks, respectively). (**A**) Anti-S IgG (positivity index) (values > 1.1 indicate the presence of antibodies in the sample); (**B**) Virus neutralization test against delta strain (titer); (**C**) IGRA (mlU/mL).

**Figure 2 viruses-14-01086-f002:**
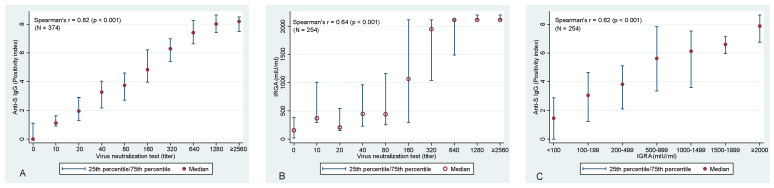
Association between (**A**) VNT and anti−S IgG antibodies, between (**B**) VNT and T−cell-mediated immunity (IRGA), and between (**C**) IGRA and anti−S IgG antibodies. Please note that 2560 is the highest used titer of VNT and, hence, it is likely that for many of these samples, an even higher titer would be detected if dilution series continued. The same applies to IGRA values with the maximum value of 2115 mlU/mL.

**Figure 3 viruses-14-01086-f003:**
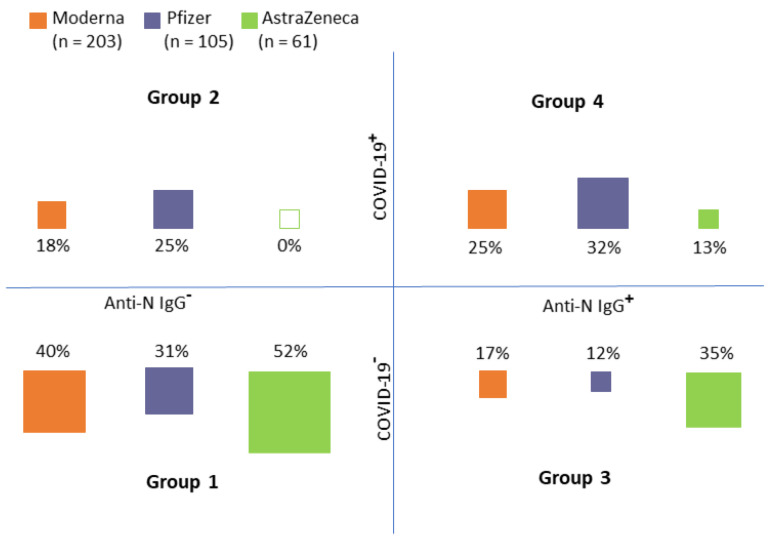
Group distribution according to the medical records of COVID−19 history (COVID−19^+^) and the presence of N−antibodies (an indicator of immune response to COVID−19 infection (Anti−N IgG^+^)) (Group 1: No COVID−19 record in personal history/no anti−N IgG, Group 2: COVID−19 record in personal history/no anti−N IgG, Group 3: No COVID−19 record in personal history/anti−N IgG positive, Group 4: COVID−19 record in personal history/anti−N IgG positive).

**Figure 4 viruses-14-01086-f004:**
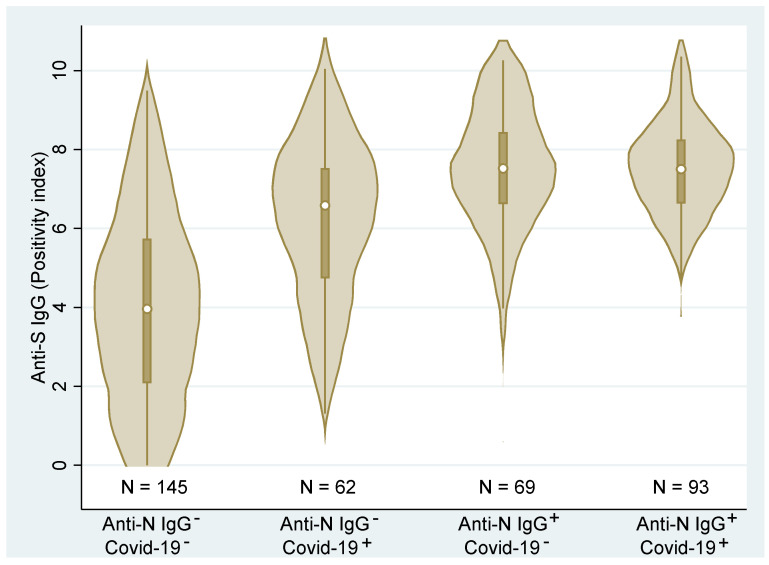
Distributions and box plots of anti−S IgG levels according to the personal COVID−19 history status based on medical records (COVID−19^−/+^) and anti−N IgG (Anti−N IgG^−/+^) (N—number of observations).

**Table 1 viruses-14-01086-t001:** Description of individual groups.

	All	Moderna	Pfizer	AstraZeneca	*p*-Value *
*n* (%)	375 (100)	209 (55.7)	105 (28.0)	61 (16.3)	<0.001
Age (mean ± SD)	80.2 ± 8.45	82.5 ± 7.97	73.8 ± 6.43	83.4 ± 7.22	<0.001 ^+^
Women (%)	258 (68.8)	159 (76.1)	56 (53.3)	43 (70.5)	<0.001
Men (%)	117 (29.5)	50 (23.9)	49 (46.7)	18 (29.5)	
Vaccination-to-test period—median (IQR), (weeks)	30 (28–31)	31 (29–32)	30 (30–30)	23 (23–23)	<0.001
N of participants with history of COVID-19 (medical records + N-antibodies)	224 (59.7)	122 (58.4)	73 (69.5)	29 (47.5)	0.017
N of participants with history of COVID-19 based on medical records (%)	155 (41.3)	87 (41.6)	60 (57.1)	8 (13.1)	<0.001
N of participants with history of COVID-19 based on N-antibodies (%)	162 (43.2)	86 (41.1)	47 (44.8)	29 (47.5)	0.628
Diabetes (%)	133 (38.9)	80 (40.4)	28 (32.2)	25 (43.9)	0.297
Cardiovascular diseases (%)	226 (66.1)	141 (71.2)	32 (36.8)	53 (93.0)	<0.001
High systolic blood pressure (%)	246 (71.9)	145 (73.2)	54 (62.1)	47 (82.5)	0.024
Chronic respiratory diseases (%)	56 (16.4)	39 (19.7)	8 (9.2)	9 (16.1)	0.088
Hematooncological disease (%)	10 (2.9)	7 (3.5)	3 (3.5)	0 (0)	0.448 ^++^
Other oncological disease (%)	31 (9.1)	17 (8.6)	7 (8.1)	7 (12.5)	0.618
Autoimmune diseases (%)	20 (5.9)	14 (7.1)	6 (6.9)	0 (0)	0.124
Obesity (BMI > 30) (%)	48 (14.1)	28 (14.1)	17 (19.5)	3 (5.4)	0.059

*n*—number of observations, SD—standard deviation, IQR—interquartile range; * chi-squared test; ^+^ Kruskal–Wallis test; ^++^ Fisher‘s exact test.

**Table 2 viruses-14-01086-t002:** Results of anti-S IgG, virus neutralization test, and results of interferon-gamma release assay indicating T-cell immunity for individual vaccines.

		Vaccine			Total
Parameter	Result—*n* (%)	Spikevax (Moderna)	Comirnaty (Pfizer)	Vaxzevria (AstraZeneca)	
Anti-S IgG	Negative	4 (1.9)	5 (4.8)	6 (9.8)	15 (4.0)
	Borderline	2 (1.0)	1 (1.0)	0	3 (0.8)
	Positive	203 (97.1)	99 (94.2)	55 (90.2)	357 (95.2)
	Total	209 (100)	105 (100)	61 (100)	375 (100)
	*p*-value *	0.056			
VNT	Negative	3 (1.4)	5 (4.8)	10 (16.4)	18 (4.8)
	Borderline	3 (1.4)	4 (3.9)	2 (3.3)	9 (2.4)
	Positive	203 (97.2)	95 (91.3)	49 (80.3)	347 (92.8)
	Total	209 (100)	104 (100)	61 (100)	374 (100)
	*p*-value *	<0.001			
IGRA	Negative	8 (6.7)	1 (1.2)	5 (10.0)	14 (5.5)
	Borderline	7 (5.8)	3 (3.6)	4 (8.0)	14 (5.5)
	Positive	105 (87.5)	80 (95.2)	41 (82.0)	226 (89.0)
	Total	120 (100)	84 (100)	50 (100)	254 (100)
	*p*-value *	0.103			

* Fisher’s exact test, Anti-S IgG, VNT—virus neutralization test, IGRA—interferon-gamma release assay indicating T-cell immunity.

**Table 3 viruses-14-01086-t003:** Comparison of the virus neutralization titer against delta and omicron strains.

		Median (IQR)		
Group	*n*	VNT Delta (Titer)	VNT Omicron (Titer)	*p*-Value *
1: No COVID-19 record in personal history/no anti-N IgG	10	80 (20–80)	20 (10–40)	0.016
2: COVID-19 record in personal history/no anti-N IgG	10	320 (320–640)	160 (80–320)	0.059
3: No COVID-19 record in personal history/anti-N IgG positive	10	1280 (640–2560)	320 (160–640)	0.008
4: COVID-19 record in personal history/anti-N IgG positive	10	640 (320–1280)	160 (80–320)	0.002
Total	40	320 (120–1280)	160 (40–320)	<0.001

* Wilcoxon signed-rank test, IQR—interquartile range, VNT—virus neutralization test, *n*—count.

## Data Availability

Data will be available from the corresponding author at reasonable request.

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
