# Peer review of "Immune Response 5–7 Months after Vaccination against SARS-CoV-2 in Elderly Nursing Home Residents in the Czech Republic: Comparison of Three Vaccines"

_viruses, 2022, doi:10.3390/v14051086_

Round 1

Reviewer 1 Report

The authors studied the immune responses in the elderly population in nursing homes 5-7 months after two doses of 3 different vaccines. Although many papers have already address the issues of antibody responses from vaccination, it is interesting to know that the authors compared both humoral and cellular immunity among elderly people in the Czech Republic.

Personally, I like the discussion part which mentioned some possible explanation of the results. 

Some suggestions to the authors:

  1. The title “Sufficient immune response….” may be not appropriate since it is hard to judge “sufficient” from the data. We can only know the patients develop immune responses. Whether the immune responses protect the nursing home residents from infection or death is not addressed in this manuscript.
  2. I would suggest the authors provide anti-S vs T cell immunity and VNT vs T cell immunity figures in Fig.2. if they have the figures.
  3. I think the population in Group 2 (COVID in medical record but no anti-N Ab) is kind of high. Although the authors guessed that the PCR tests may amplify the virus debris, the insufficient virus may not induce immune responses. Do they find similar reports in literature?

Author Response

Dear reviewer, thank you very much for your comments. We prepared revisions accoding them. 

1. The title “Sufficient immune response….” may be not appropriate since it is hard to judge “sufficient” from the data. We can only know the patients develop immune responses. Whether the immune responses protect the nursing home residents from infection or death is not addressed in this manuscript.

This is a fair point. We removed the word "Sufficient" from the title.

Immune Response 5-7 Months After Vaccination against SARS-CoV-2 in the Elderly Nursing Homes Residents in the Czech Republic: Comparison of three vaccines

2. I would suggest the authors provide anti-S vs T cell immunity and VNT vs T cell immunity figures in Fig.2. if they have the figures.

We have added the additional graphs as suggested.

3. I think the population in Group 2 (COVID in medical record but no anti-N Ab) is kind of high. Although the authors guessed that the PCR tests may amplify the virus debris, the insufficient virus may not induce immune responses. Do they find similar reports in literature?

It is necessary to highlight the fact that all patients in our group were vaccinated against SARS-CoV-2. Hence, the finding that they are positive for S-protein, VNT and T-cell immunity but have no N-protein is indeed consistent with false-positive PCR test. An alternative explanation is offered by VanElslande et al. (doi:10.1016/j.diagmicrobio.2022.115659) who reported that anti-N-protein antibodies persist in the organism for a shorter time than anti-S protein. (references 4,15, 19-24) Also, patients with a milder course of infection have lower initial levels of anti-N antibodies as well, making its decline all the more likely.

Best regards

Hana Tomaskova

Reviewer 2 Report

The study is well conducted with high number of subjects recruited. There is indeed the limitation of comparing Pfizer and Moderna at 7 months versus AstraZeneca at 5 months but it is highlighted and the literature is in favour of the conclusions drawn here. 

The other limitation (in the context of delta and omicron) in my opinion still coule be explored to give more power to the study. Even if the occurence of infection by these variants was not feasible I assume the immune repose (level of plasma antibodies, neutralisation) can be assessed in the samples collected before the booster (basically the samples used for this study) by using delta and omicron strains. This is of high interest since the general conclusion is focuses on the requirement of the booster, which should be evaluate in the most current context (so at least against Omicron).

In addition, it is well highlighted in the introduction that even if the antibodies in the plasma are waining, memory B cells can persist on the long term and secrete antibodies if re-activated. Why not looking at memory B cells here? 

Author Response

Dear reviewers,

thank you for your commets. We prepared the revisions and the new analysis of samples accroding them.  

1. The other limitation (in the context of delta and omicron) in my opinion still coule be explored to give more power to the study. Even if the occurence of infection by these variants was not feasible I assume the immune repose (level of plasma antibodies, neutralisation) can be assessed in the samples collected before the booster (basically the samples used for this study) by using delta and omicron strains. This is of high interest since the general conclusion is focuses on the requirement of the booster, which should be evaluate in the most current context (so at least against Omicron).

This is a fair point. We performed an additional virus neutralization test with omicron in a subset of 40 samples (10 samples randomly chosen from each of the four groups). The results confirmed a good qualitative correlation between delta and omicron neutralizing capacity; however, the median neutralization capacity against omicron was about half that against delta. We have added this analysis and information into the manuscript.

2. In addition, it is well highlighted in the introduction that even if the antibodies in the plasma are waining, memory B cells can persist on the long term and secrete antibodies if re-activated. Why not looking at memory B cells here? 

It is because we do not have the method for B-cells available in our laboratory. Analyzing the activity of B-cels is much more complicated, among other things necessitating the use of fresh blood. For this reason, we used humoral immunity (VNT and anti-S+anti-N antibodies) and cellular immunity (T-cell response, IGRA). Studies for B-cell immunity can be found, e.g., here: Kang CK et al. 2022. doi:10.3389/fimmu.2022.830433, Gaebler et al. 2021, doi: 10.1038/S41586-021-03207-W.). We have not this data.

Best regards

Hana Tomaskova

Round 2

Reviewer 2 Report

The authors took in consideration the comment about the importance of including data on variants (delta and omicron), they made the effort to run neutralisation assays and include the conclusions in the reviewed manuscript. These new analysis bring more power to the overall study, increase the interest for the readers and thus justify my positive recommendation for publication.